# Emergent Communication for Rules Reasoning

**Yuxuan Guo**[1,2,3]    **Yifan Hao**[2]    **Rui Zhang**[2]    **Enshuai Zhou**[1,2,3]    **Zidong Du**[2,5]
**Xishan Zhang**[2,3]    **Xinkai Song**[2]    **Yuanbo Wen**[2]    **Yongwei Zhao**[2]    **Xuehai Zhou**[1]
**Jiaming Guo**[2]    **Qi Yi**[1,2,3]    **Shaohui Peng**[6]    **Di Huang**[2]    **Ruizhi Chen**[6]
**Qi Guo**[2]    **Yunji Chen**[2,4] [*]

[1]University of Science and Technology of China
[2]State Key Lab of Processors, Institute of Computing Technology, CAS
[3]Cambricon Technologies    [4]University of Chinese Academy of Sciences
[5]Shanghai Innovation Center for Processor Technologies
[6]Intelligent Software Research Center, Institute of Software, CAS

gyx_20170818@mail.ustc.edu.cn, {haoyifan, cyj}@ict.ac.cn

## Abstract

Research on emergent communication between deep-learning-based agents has received extensive attention due to its inspiration for linguistics and artificial intelligence. However, previous attempts have hovered around emerging communication under perception-oriented environmental settings, that forces agents to describe low-level perceptual features intra image or symbol contexts. In this work, inspired by the classic human reasoning test (namely Raven's Progressive Matrix), we propose the Reasoning Game, a cognition-oriented environment that encourages agents to reason and communicate high-level rules, rather than perceived low-level contexts. Moreover, we propose 1) an unbiased dataset (namely rule-RAVEN) as a benchmark to avoid overfitting, 2) and a two-stage curriculum agent training method as a baseline for more stable convergence in the Reasoning Game, where contexts and semantics are bilaterally drifting. Experimental results show that, in the Reasoning Game, a semantically stable and compositional language emerges to solve reasoning problems. The emerged language helps agents apply the extracted rules to the generalization of unseen context attributes, and to the transfer between different context attributes or even tasks.

## 1    Introduction

Research on emergent communication has received extensive attention in recent years due to its inspiration to the fields of linguistics and artificial intelligence (AI). From the linguistic perspective, the study of emergent communication may help understand the origin and evolution of natural language [9, 13, 14, 28, 40]. From the AI perspective, introducing communication mechanism into the deep-learning-based multi-agents system is a promising way to make it interpretable [11], human-interactable [25], and more importantly, generalizable [1, 7, 31, 37].

Lewis-style Game [27] is widely used in previous studies to simulate emergent communication between two agents, a speaker, and a listener. In this game, the speaker sends a message based on perceived contexts (usually images or symbols), and the listener completes the perception task (usually discriminating or reconstructing the target object) based on the message. As two agents

---

[*]Corresponding author.

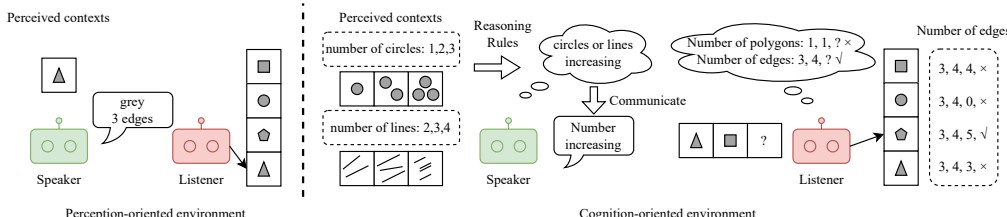

Figure 1: Perception-oriented environment (Left, i.e., *speak by seeing*) v.s. cognition-oriented environment (Right, i.e., *speak by reasoning*) for emergent communication.

gradually endow messages with precise semantics to describe the target object more accurately, a machine language emerges with communication.

However, previous attempts have hovered around emerging communication under perception-oriented environmental settings, that forcing agents to describe low-level perceptual features (e.g., shape, color and combinations of them) intra image [11, 14, 15, 23, 24, 31] or symbol [28, 35–37] contexts. Unfortunately, such perception-oriented environments (i.e., *speak by seeing*) are unable to prompt agents to emerge cognitive-based communication (i.e., *speak by reasoning*), which is considered the foundation of human language and intelligence evolution by research in linguistics [12] and cognitive psychology [6]. Therefore, how to emerge communication from perception to cognition remains an unsolved issue.

In this work, we propose the Reasoning Game, a cognition-oriented environment that encourages agents to reason and communicate high-level rules, rather than perceived low-level contexts. Figure 1 shows the comparison between the perception-oriented environment and cognition-oriented environment. The Reasoning Game requires two agents to cooperatively solve a classic human reasoning test Raven Progressive Matrix [34] (RPM). Specifically, the speaker reasons underlying rules from two rows of contexts (consisting of visually simple attributes), and sends a message to the listener. Meanwhile, the listener must correctly parse the received message and grasp the extracted rules, so as to select a candidate option that matches the rules (and fills the blank in the last row of the contexts matrix). In the process of solving RPM by agents, the rules communicated are not related to perceived features intra a single context but rather stem from abstract and structural logical relationships inter multiple contexts [39], such as gradually deepening colors and increasing numbers. According to this characteristic of RPM, agents can only complete the Reasoning Game by reasoning implicit rules at the cognitive level, instead of explicit visual features at the perceptual level. Such cognition-oriented environmental settings are considered to characterize the defining feature of high-level human-like intelligence [6, 18, 48].

Moreover, to conveniently evaluate the cognitive ability of agents through the Reasoning Game, we propose 1) an unbiased rule-RAVEN dataset as a benchmark, and 2) a two-stage curriculum agent training method as a baseline. For the former, the rule-RAVEN dataset contains numerous RPM problems with different combinations of rules, and each problem consists of context panels with multiple attributes. The dataset is debiased by our rule-based candidate panel generation algorithm to avoid overfitting in agent communication. For the latter, the two-stage curriculum agent training method ensures agents converge more stably in the Reasoning Game, where contexts and semantics are bilaterally drifting.

We summarize the contributions of this paper with experimental results:

- (Sec 5.1) The Reasoning Game successfully enables the emergent communication for reasoning rules between agents, and the emerged language is semantically stable and compositional.

- (Sec 5.2) The proposed benchmark (i.e., the rule-RAVEN dataset) and baseline (i.e., the two-stage curriculum agent training method) in the Reasoning Game are effective as expected and necessary for agent evaluation.

- (Sec 5.3) We verify the improvement of agent cognitive ability through the emergent communication for reasoning rules from three aspects: 1) generalize to unseen rule combinations, 2) apply the extracted rules to other context attributes, 3) and transfer to different tasks.

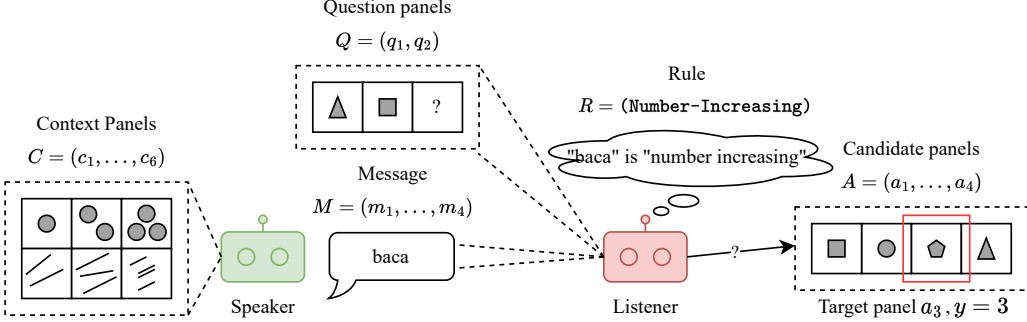

Figure 2: The Reasoning Game

## 2   Related Works

**Emergent Communication.** Using computational models to simulate emergent communication has a long history [21, 22, 43] and has recently been revived with the popularity of deep learning. A. Lazaridou et al. [24] studied emergent communication on the basis of reinforcement learning and deep neural network agents for the first time. Starting from this, dozens of studies focus on exploring the properties of the emerged language from the perspective of linguistics and cognitive psychology, such as alignment of agents [11], efficiency of language [8], capacity of communication channel [23], community interaction [15], ease of teaching [28], culture transmission [14], population heterogeneity [31], and environment scale [10]. In general, these studies adopt perception-oriented environmental settings, restricting the emerged languages to describing low-level perceptual features intra image or symbol contexts. Specifically, previous works [11, 15, 24] emerge communication of intra-context object attributes (e.g., color, shape) and their combinations (e.g., `blue OR/AND triangle`) [31]. On the contrary, our work focuses on emergent communication in a cognition-oriented environment (i.e., Reasoning Game), where agents reason inter-context high-level rules and apply them, so as to cooperatively solve reasoning tasks (i.e., RPM).

**Raven's Progressive Matrices (RPM).** RPM [34] is a widely studied human IQ test, which has recently been revived in the AI community after it can be procedurally generated [44]. Compared with other reasoning benchmarks, such as Visual Question Answering [4, 19, 29] and Arithmetic Reasoning [38], RPM focuses on relational reasoning inter multiple contexts, reflecting humans' core cognitive abilities [6]. Based on RPM, many synthetic [2, 48] and natural [41] datasets and model designs [17, 46] are proposed to boost the high-level reasoning capability of deep learning models. In this work, we are the first to study emergent communication through cooperatively solving RPM problems. We find that such communication scenarios need to ensure clear semantics and avoid overfitting so that imposing new requirements on the rules set design and candidate panels generation of the RPM dataset. Motivated by this, we propose the rule-RAVEN dataset as a benchmark, and its corresponding two-stage curriculum agent training method as a baseline.

## 3   Method

### 3.1   Reasoning Game

The Reasoning Game is a cognition-oriented environment that requires two agents to solve RPM problems by reasoning and communicating high-level rules. The game can be formalized as a 5-tuple: $(C, Q, A, R, y)$, where $C = (c_1, \ldots, c_l)$ is a set of $l$ context panels, $Q = (q_1, \ldots, q_m)$ is a set of $m$ question panels, $A = (a_1, \ldots, a_n)$ is a set of $n$ candidate panels with 1 target panel and $n-1$ distract panels, $R = (r_1, \ldots, r_k)$ is a vector that encodes the rules implicit in $C$ (i.e., hidden to the speaker and listener), and $y \in \{1, \ldots, n\}$ indicates the index of the target panel, the only panel together with the question panels that match the rule $R$. Each panel contains several attributes (such as number, color, et al.) $T = (t_1, \ldots, t_k)$, and the values of all attributes are within a pre-defined range.

Figure 2 shows the scheme of the Reasoning Game. In this scheme, the speaker first reasons the embedding, which represents the rules $R$ implicit in context panels $C$, then encodes the embedding

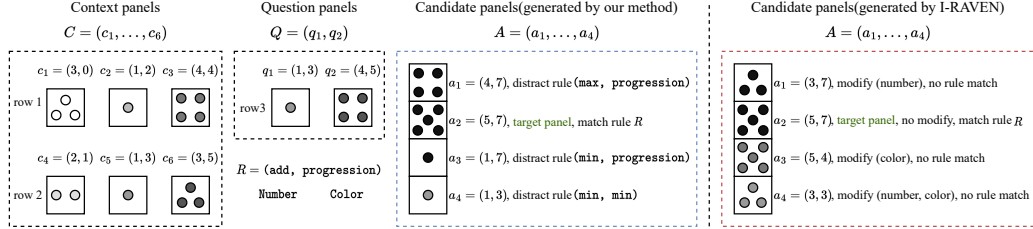

Figure 3: (*left*) A reasoning problem example in the rule-RAVEN dataset, where the candidate panels are generated by introducing distract rules, shown in Algorithm 1. (*right*) Previous RPM-based datasets (e.g., I-RAVEN) generate candidates by hierarchically modifying some attribute values of the target panel, making it easy for the listener to distinguish without considering communication from the speaker.

into a message $M = (m_1, \ldots, m_s)$ (each token $m_i$ is derived from a pre-defined vocabulary $V$) for sending to the listener. Finally, according to the received message $M$, the listener chooses the target panel from candidate panels $A$ to fill the blank behind the question panel $Q$. If the reasoning question is correctly solved, the listener and the speaker will be rewarded simultaneously.

## 3.2   Rule-RAVEN dataset

A reasoning problem (e.g., RPM) can be naturally divided into two steps: extracting rules and applying rules. It seems that the reasoning dataset (e.g., RAVEN) can be straightforwardly used to play the Reasoning Game by assigning the former step (i.e., extracting rules) to the speaker and the latter step (i.e., applying rules) to the listener. However, the premise for such a scheme of Reasoning Game could proceed successfully is that the speaker and listener can effectively transmit and understand the extracted rules by emergent communication. To this end, crafting a reasoning dataset as a benchmark still has requirements on the rules set design and candidate panels generation.

**Regarding the rules set design**, the mapping relationship between implicit rules and context attributes must be unambiguous to emerge semantically clear communication and avoid imbalanced prior probability distribution of rules' frequency in the dataset. Specifically, the design principle is subdivided into two aspects: 1) The rule corresponding to each set of attribute values in panels is unique. For example, if the attribute (e.g., size) value in context panels $(c_1, c_2, c_3)$ is $(1, 2, 3)$, which matches both rules 'sum' ($1 + 2 = 3$) and 'progression +1' ($1 + 1 = 2, 2 + 1 = 3$), then the attribute value should be filtered out from the dataset. 2) The meanings expressed between different rules do not overlap. For example, when any set of attribute value $(a, b, c)$ matches the rule 'progression' ($a + const = b, b + const = c$), it also matches the rule 'average' ($b = \frac{a+c}{2}$). In this case, only one of the rules should be retained in the dataset.

**Regarding the candidate panels generation**, distract panels should be generated carefully so as to represent close but not quite correct rules that force the speaker and listener to communicate precisely the correct rules. Specifically, the design principle is that each distractor in candidate panels should match a 'distract rule' (as shown in figure 3 'our method'), to prevent the listener from inferring the target solely through the existence of regular rules between the question panels and candidate panels (a negative example is shown in figure 3 (*right*) 'I-RAVEN [17]' ). Otherwise, it will cause the speaker to lose the motivation to extract and communicate rules correctly (we will demonstrate this in detail by experiments in Sec 5.2).

Following the above requirements, we propose a **symbolic dataset** [2], rule-RAVEN, for the Reasoning Game. Figure 3 (*left*) shows an example case in the rule-RAVEN dataset (See Appendix A for more details.).

## 3.3   Model and Training

We build the speaker's and listener's model in the Reasoning Game as follows.

---

[2]The rule-RAVEN dataset is based on symbols rather than images, mainly because visual recognizability limits the attribute values.

**Algorithm 1:** Rule-based candidate panels generation algorithm

**Input:** $Q$, $R$, distract_panels_num
**Output:** distract_panels
// generate distract rule space

1  possible_rules = []
2  **for** *i=0 to $|R| - 1$* **do**
3     tmp_rule = []
4     **for** *r ∈ rule_set* **do**
5         **if** *check_satisfy(Q[:, i], r)* **then**
6             tmp_rule.append(r)
7         **end**
8     **end**
9     possible_rules.append(tmp_rule)
10  **end**
11  distract_rule_space = cartesian_product(possible_rules) - R
    // generate distract panels
12  distract_panels = []
13  **for** *rs ∈ random_sample(distract_rule_space, distract_panels_num)* **do**
14     tmp_panel = []
15     **for** *i=0 to $|rs| - 1$* **do**
16         tmp_attr = reason(rs[i], Q[:, i])
17         tmp_panel.append(tmp_attr)
18     **end**
19     distract_panels.append(tmp_panel)
20  **end**
21  return distract_panels

**Speaker's model.** The speaker consumes context panels $C$ and outputs message $M$. The speaker's policy can be formalized by $S(M|C; \theta)$ with trainable parameters $\theta$. Given context panels $C$, an MLP-based perception module $f^S$ first encodes each panel into an embedding $(f^S(c_i))_{i=1}^6$. Then, a reasoning module $g^S$ extracts the row-level rule embedding $r_1^S = g^S((f^S(c_i))_{i=1}^3)$ and $r_2^S = g^S((f^S(c_i))_{i=4}^6)$ from multiple context embeddings. We follow the state-of-art RAVEN solver SCL [46]'s core module, shared group MLPs, as the reasoning module. The row-level rule embeddings are average-aggregated to get the rule embedding $r^S = mean(r_1^S, r_2^S)$ of context panels $C$. Finally, an LSTM [16] message encoder $h^S$ takes the rule embedding $r^S$ as initial state $z_0^S$ and a special token $m_0 = $ <SOS> as start token to generate message $M$. At each timestep $t \geq 1$, based on current state $z_t = h^S(z_{t-1}, m_{t-1})$, message $m_t = sample(softmax(q(z_t)))$, where $q$ is a linear transformation to project the hidden state space to the vocabulary space. Token generation terminates until the message reaches a fixed length or a special token <EOS> is generated.

**Listener's model.** The listener consumes message $M$, question panels $Q$, and candidate panels $A$ and outputs a prediction $\hat{y}$. The listener's policy can be formalized by $L(\hat{y}|M, Q, A; \phi)$ with trainable parameters $\phi$. Given question panels $Q$ and candidate panels $A$, a perceptual module $f^L$ first encodes $Q$ and $A$ into corresponding embeddings $(f^L(q_i))_{i=1}^2$ and $(f^L(a_i))_{i=1}^8$. Meanwhile, an LSTM message decoder $h^L$ takes a special state $z_0^L$ as the start state to decode message $M$. At each timestep $t \geq 1$, current state $z_t^L = h^L(z_{t-1}^L, m_t)$. Message decoding terminates until it reaches the maximum message length $|M|$ or <EOS>. We use the final state $z^L$ as the decoded message embedding, representing the rules implied in context panels $C$. Then, each candidate embedding is concatenated with the question embedding and fed into the reasoning module $g^L$ to obtain multiple candidate rule embeddings $r_i^L = g^L(f^L(q_1), f^L(q_2), f^L(a_i)), i \in \{1, \ldots, 8\}$. Finally, the message embedding and the candidate rule embedding perform an inner product and softmax operation to obtain the probability distribution $softmax((r_i' \cdot z_{|M|}^T)_{i=1}^8)$ of candidate panels, which is further used to compute the loss and the prediction $\hat{y}$.

To ensure that the cognitive capabilities of the speaker and the listener are similar, we build their perception and reasoning module architecture identically. No parameters are shared between the

speaker and listener to avoid introducing artificial priors about their consensus. More details of the model can be found in Appendix B.

In our game, both rules reasoning and language emergence are highly challenging. We find that jointly training the speaker and listener directly from scratch leads to communication failures. The reason is that, in the early stage of joint training, the reasoning context panels and the message semantics are drifting simultaneously, leading the speaker cannot produce helpful messages to the listener, and the listener tends to ignore the speaker's message. To help agents jump out of the local optimum, we propose a two-stage curriculum training [3] method.

**In the first stage,** the speaker is pre-trained on a held-out rule set (See Appendix A for more details.) generated dataset to acquire some reasoning capabilities to cope with the drifting context. This stage can be considered a supervised learning task with the ground-truth rule vector $R$ as the label and only updating the speaker's parameters $\theta$. Note that this does not impose artificial priors on emerged language since the rule encodings predicted at this stage are absent in the joint training stage. The gradients of the objective function $J_1$ in the first stage are:

$$\nabla_\theta J_1 = \mathbb{E}[\sum_{i=1}^{k} \mathbb{1}(r_i, m_i)\nabla \log S(m_i|C)], r_i \in R, m_i \in M$$

Note that although the $\mathbb{1}(\cdot, \cdot)$ operator exists in $\nabla_\theta J_1$, this does not introduce constraints on the size of emerged language (i.e., `message_length` and `vocabulary_size`) in the second stage of training. See Appendix C for more details.

**In the second stage,** the pre-trained speaker and listener train jointly to emerge language. Since discrete communication breaks the end-to-end differentiability, we employ the REINFORCE[45] algorithm. The gradients of the objective function $J_2$ in the second stage are:

$$\nabla_\theta J_2 = \mathbb{E}[\mathbb{1}(\hat{y}, y)\nabla \log S(M|C)] + \lambda \cdot \nabla H[S(M|C)]$$
$$\nabla_\phi J_2 = \mathbb{E}[\mathbb{1}(\hat{y}, y)\nabla \log L(\hat{y}|M, Q, A)]$$

Where $\lambda$ is a hyperparameter controlling entropy regularization.

## 4 Exprimental Settings

### 4.1 Basic Settings

**Implementation.** We use Python3 [42] to implement the rule-RAVEN dataset. Our model implementation is based on Pytorch [33] and EGG [20] toolkit. The code is available in supplementary materials.

**Dataset preparation.** We instantiate our rule-RAVEN dataset (introduced in Sec 3.2). For each case of reasoning problem, there are 6 context panels (i.e., $C = (c_1, \ldots, c_6)$), 2 question panels (i.e., $Q = (q_1, q_2)$), and 8 cadidate panels (i.e., $A = (a_1, \ldots, a_8)$). Each panel has 4 attributes (i.e., $T = (number, color, size, shape)$), and each attribute have $N \in \{20, 30, 40\}$ values ($N^4$ in total). The larger $N$ (more diverse the attribute values) means it is more difficult to identify contexts and reason rules. There are 8 rules that apply to each attribute respectively, resulting in a total of $8^4 = 4096$ different rule combinations. For each rule combination, We randomly generated 20 different reasoning problem cases (sampling from $N^4$ attribute values) with Algorithm 1 ($4096 \times 20 = 81920$ in total), and half of these cases for training, half for testing. Besides, we use a held-out rule set to additionally generate 2000 cases for pre-training the speaker in the first stage of agent training (introduced in Sec 3.3).

**Communication Channel.** We set the maximum message length $|M| = 4$ and the vocabulary size $|V| = 15$. With this setup, the capacity of the communication channel is $15^4$, sufficient to describe all rule combinations ($> 8^4$), and preserve redundant tokens for two-stage curriculum training, and much lower than vocabularies required to directly describe the context panels ($\sim 10^{10}$).

**Optimization.** Agents' parameters are optimized by AdamW [30], with a learning rage of $3 \cdot 10^{-3}$, a weight decay of $0.01$, $\beta_1 = 0.99$, $\beta_2 = 0.999$ and a batch size of 512. For the speaker, we set the hyperparameter of entropy regularization $\lambda = 0.01$.

## 4.2 Language Metrics of emergent communication

**Generalization abilities.** To test the generalization ability of the emerged language, we further create 4 data splits, corresponding to 4 levels of generalization ability: 1) In-distribution generalization (`ID`). The training and test sets share the same rule combinations but consist of different problems. The `ID` level tests the language's ability to generalize to different problems with the same rules. 2) Interpolated out-of-distribution generalization (`Inpo-ood`). Each rule in the rule set appears in each attribute on the train set, but some rule combinations do not appear on the train set and form the test set. The `Inpo-ood` level tests the language's ability to generalize to unseen rule combinations based on components. 3) Extrapolated out-of-distribution generalization level-1 (`Expo-ood-L1`). Each attribute has a unique rule that does not appear in the train set, and these rules are combined with other rules that appear in the train set to form a test set. The `Expo-ood-L1` level tests the language's ability to describe rules across attributes. 4) Extrapolated out-of-distribution generalization level-2 (`Expo-ood-L2`). A rule does not appear on any attribute on the train set, and it is combined with other rules present in the train set to form the test set. The `Expo-ood-L2` level tests the language's ability to generalize to new rules.

**Topographic similarity (Topsim).** Topsim [5] is widely used in previous emergent language studies to measure language compositionality. It is defined as the Spearman correlation coefficient of pairwise distances between input and message space. In the input space, we use the normalized Hamming distance between rules and the cosine distance between panels. In the message space, we adopt the Levenshtein [26] distance with the same removal, insertion, and substitution costs. Computing Topsim involves pairwise distances and thus has a computational complexity of $O(n^2)$, which is computationally prohibitive in our environment. In the experiment, we randomly sample 1000 samples per run to calculate Topsim and take the average over 20 runs as the final result.

**Ease and transfer learning (ETL).** ETL [10] captures how fast and well the emergent language is transmitted to new listeners performing a different task. We focus on the emerged language's transferability between reasoning tasks with different attribute values, $N$. Specifically, we first dump the deterministic languages (choosing the token greedily with the highest probability) of well-trained speakers under each attribute value setting. We then map deterministic languages in the source task setting and reasoning problems in the target task settings based on the rules. Finally, we train new randomly initialized listeners using mapped language and reasoning problems and observe the convergence speed and accuracy of the listeners. Our source task setting contains attribute values $N \in \{20, 30, 40\}$, and the target task setting contains $N \in \{20, 30, 40\}$, and an extremely large $N = 80$.

# 5 Exprimental Results

To show the benefits and effectiveness of emergent communication through the Reasoning Game, we answer the following three questions by the experimental results.

## 5.1 What do the emerged language tend to describe?

**Emerged language prefers to describe rules rather than context panels.** We demonstrate this issue from two aspects. 1) Compare which input (rules or context panels) has a stronger correlation (measured by *Topsim*) with the message space. As Table 1 shows, in all three diversities of attribute values (i.e., $N = 20, 30, 40$), the *Topsim* between rules and messages (i.e., `Rule-Message`) is always substantially higher than that between context panels and messages (i.e., `Panel-Message`). This directly proves the conclusion, and the emerged languages are somewhat compositional (because *Topsim* is also used to measure compositionality). 2) Another indirect evidence comes from the relationship between ID generalization performance (measured by Listeners' accuracy) and attribute value diversities. Specifically, since the capacity of the communication channel is not enough to describe the context panels ($15^4 < 10^{10}$), more diverse context panels are expected to exacerbate this shortage, leading to a rapid decrease in accuracy. However, paradoxically, the data in each column of Table 2 shows that the accuracy is almost equal as the attribute value diversity increases (from 20 to 30 or 40).

Table 1: *Topsim* (mean $\pm$ standard error) between rules/panels and messages on `ID` data split with p-value$< 10^{-4}$.

| Attribute values | Rule-Message | Panel-Message |
|---|---|---|
| 20 | $0.2382 \pm 0.0117$ | $0.1148 \pm 0.0122$ |
| 30 | $0.2540 \pm 0.0130$ | $0.1127 \pm 0.0080$ |
| 40 | $0.2381 \pm 0.0289$ | $0.0937 \pm 0.0124$ |

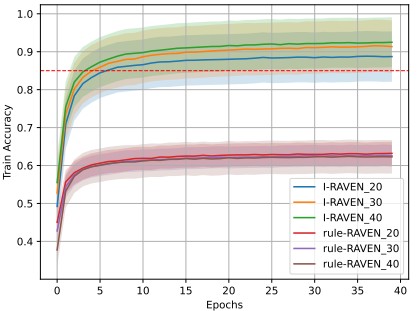

Figure 4: Train accuracy under the *message-blocked* scene. The results are averaged across the performance of 16 randomly initialized listeners on 4 randomly generated dataset instances. The shaded region represents the standard deviation.

Figure 5: Train accuracy w/ and w/o two-stage curriculum training. The results are averaged across 8 seeds. The shaded region represents the standard deviation.

## 5.2 Why are the proposed benchmark and baseline effective and necessary?

**Rule-RAVEN dataset.** We quantitatively compare our rule-based candidate panels generation algorithm (i.e., Algorithm 1) in rule-RAVEN with the I-RAVEN. Specifically, by separately training the listener under a *message-blocked* scene (i.e., the speaker's message is forced to a constant), we compared the overfitting degree of rule RAVEN and I-RAVEN (the higher the accuracy, the less dependent the listener is on the speaker's reasoning about rules, i.e., the higher the overfitting degree). Figure 4 shows that I-RAVEN suffers from overfitting severely (lines with the legend 'I-RAVEN_20/30/40'), for the listener's train accuracy still achieves $\sim 0.9$ even if the speaker's message is completely ignored. While rule-RAVEN (lines with the legend 'rule-RAVEN_20/30/40') can effectively control the degree of overfitting, the listener's train accuracy is only $\sim 0.6$, which is significantly different from when receiving the speaker's message normally ($\sim 0.6$ v.s. $\sim 0.9$). So it is enough to distinguish whether communication is effective or not. Note that controlling overfitting to an ideal level (8 candidate panels, $\approx 0.125$ message-free accuracy) is unlikely because the listener can always guess the answer by memorizing the patterns corresponding to specific rules. Besides, naive methods, such as increasing attribute values, do not alleviate overfitting (as shown in lines with the legend 'I-RAVEN,' from 20 to 30 or 40).

**Two-stage curriculum training method.** We compare the training curves with and without the two-stage curriculum training method. Figure 5 shows that the two-stage curriculum training method leads to successful communication in all three attribute value diversities (lines with the legend 'w_pt_20/30/40'), for agents reaches higher accuracy ($\sim 0.9$). In contrast, in the case without the two-stage curriculum training method (lines with the legend 'w/o_pt_20/30/40'), agents only converge to a similarly low accuracy as under the *message-blocked* scene, which implies that agents can not communicate successfully. An intuitive explanation for this phenomenon is that, the speaker's training in the first stage helps him acquire some reasoning ability to cope with the drifting context, which helps stabilize the semantic production in the joint training stage.

Table 2: Accuracy (mean ± standard error) of 4 generation levels.

| Attribute values | ID | Inpo-ood | Expo-ood-L1 | Expo-ood-L2 |
|---|---|---|---|---|
| 20 | $0.9492 \pm 0.0034$ | $0.9473 \pm 0.0039$ | $0.6590 \pm 0.0371$ | $0.4921 \pm 0.0658$ |
| 30 | $0.9369 \pm 0.0073$ | $0.9295 \pm 0.0083$ | $0.5880 \pm 0.0349$ | $0.4684 \pm 0.0630$ |
| 40 | $0.9194 \pm 0.0190$ | $0.9210 \pm 0.0178$ | $0.5741 \pm 0.0238$ | $0.4597 \pm 0.0379$ |

Table 3: Train accuracy (mean ± standard error) of task transferring.

| | | Target | | | |
|---|---|---|---|---|---|
| | | 20 | 30 | 40 | 80 |
| Source | 20 | - | $0.9257 \pm 0.0059$ | $0.9238 \pm 0.0051$ | $0.9199 \pm 0.0081$ |
| | 30 | $0.9098 \pm 0.0110$ | - | $0.9124 \pm 0.0087$ | $0.9096 \pm 0.0083$ |
| | 40 | $0.9022 \pm 0.0171$ | $0.9043 \pm 0.0159$ | - | $0.8974 \pm 0.0136$ |

## 5.3 What are the benefits of communicating extracted rules?

**Beneficial for generalizing to unseen rule combinations.** As shown in Table 2, the high generalization performance at the ID level($\approx 0.92$) shows that communicating abstract rules can generalize to different reasoning problems with the same rule combination. Moreover, the generalization performance of the Inpo-ood level is similar to that of the ID level, which indicates that communicating abstract rules can also generalize to unseen rule combinations, e.g., based on (add, min) and (progression, max) on (*number*, *size*) generalize to (add, max) on (*number*, *size*).

**Beneficial for agents apply rules to other attributes.** As shown in Table 2, we observe that the performance of level Expo-ood-L1 is higher than that of Expo-ood-L2($\approx +0.13$), which suggests that communicating abstract rules can help the listener to apply rules extracted from one attribute to others, e.g., applying rule (progression) extracted from (*number*) to (*size*). This characteristic is similar to the human ability to draw inferences from one instance to another. Similar to the observation in Chaabouni et al. [10], complicating the task of language emergence (reasoning task in our work) does not improve the systematicness of emergent language (from average Inpo-ood$\approx 0.92$, to Expo-ood-L1 $\approx 0.58$, and to Expo-ood-L2 $\approx 0.46$), possibly because reasoning about unseen rules is highly challenging in our game.

**Beneficial for transferring to a different task setup.** As shown in Table 3, all transfers in the table have relatively high train accuracy and are close to the upper bound (ID and Inpo-ood accuracy). Even languages that emerged on the simplest task (20 values of each attribute) can be transferred to the most challenging task (80 values of each attribute). Figure 6 shows the training curve of task transferring grouped by target attribute values. Emerged language (lines with the legend 'agent') has a similar convergence speed compared with the ideal language that directly uses rule encoding as the language (lines with the legend 'rule') and only hundreds of optimization steps (less than 10 epochs) to converge. Besides, emerged language outperformed the random group (lines with the legend 'random'), which randomly maps source languages and the target reasoning problems. Good cross-task transferability also indirectly confirms that the emerged languages tend to describe rules more than context panels.

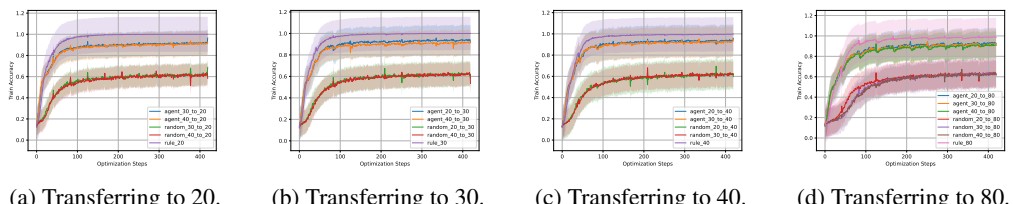

| (a) Transferring to 20. | (b) Transferring to 30. | (c) Transferring to 40. | (d) Transferring to 80. |

Figure 6: Training curve of task transferring grouped by target attribute values. The results are averaged across 8 seeds, and the shaded region represents the standard deviation.

## 6 Discussion and Conclusion

In this work, we propose the Reasoning Game, a cognition-oriented environment that encourages agents to reason and communicate high-level rules. Based on our rule-RAVEN dataset and two-stage curriculum agent training method, we successfully observe the emergence of a language that can describe abstract rules and further reveal the language's impressive generalization ability and transferability.

Our research may inspire the following directions: 1) Emerging communication on more realistic-stimuli-based reasoning datasets [32, 41, 47] to study the co-evolution of perception, cognition, and language system in agents; 2) Debiasing and avoiding overfitting on the procedurally generated reasoning dataset from a more fine-grained perspective; 3) Designing pre-training tasks to converge in environments with multiple variable drifts.

## 7 Limitations

We summarized the limitation of our works as follow: 1) Our work only focuses on language emergence on a clean symbolic-based reasoning dataset, lacking the exploration of more realistic stimuli-based (e.g., synthetic or natural images) reasoning datasets; 2) The reasoning task (RAVEN) adopted in our work only requires the agent to complete the reasoning via a single round of interaction, simplifying the natural reasoning process; 3) Our work only analyzes the semantics of the emerged languages at the message level, lacking fine-grained structural (e.g., gramma) and semantic analysis at the token level (e.g., the degree of polysemy and ambiguity between tokens).

## Acknowledgments and Disclosure of Funding

This work is partially supported by the NSF of China (under Grants 61925208, U22A2028, 62102399, 62222214, 62102398, U19B2019), CAS Project for Young Scientists in Basic Research (YSBR-029), Youth Innovation Promotion Association CAS and Xplore Prize.

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

# A Details of rule-RAVEN dataset

## A.1 Rules in rule-RAVEN

In the joint training stage, a specific rule contains 4 elements on different attributes, e.g., (Number:add, Color:progression_2, Size:constant, Shape:max). These rules meet the structural and functional requirements stated in Section 3.2 The embodiment of each rule on each attribute are shown in Figure 7. For example, the 'Color:progression_2' means: in the same row of panels, the image color gradually darkens by 2 degrees from left to right.

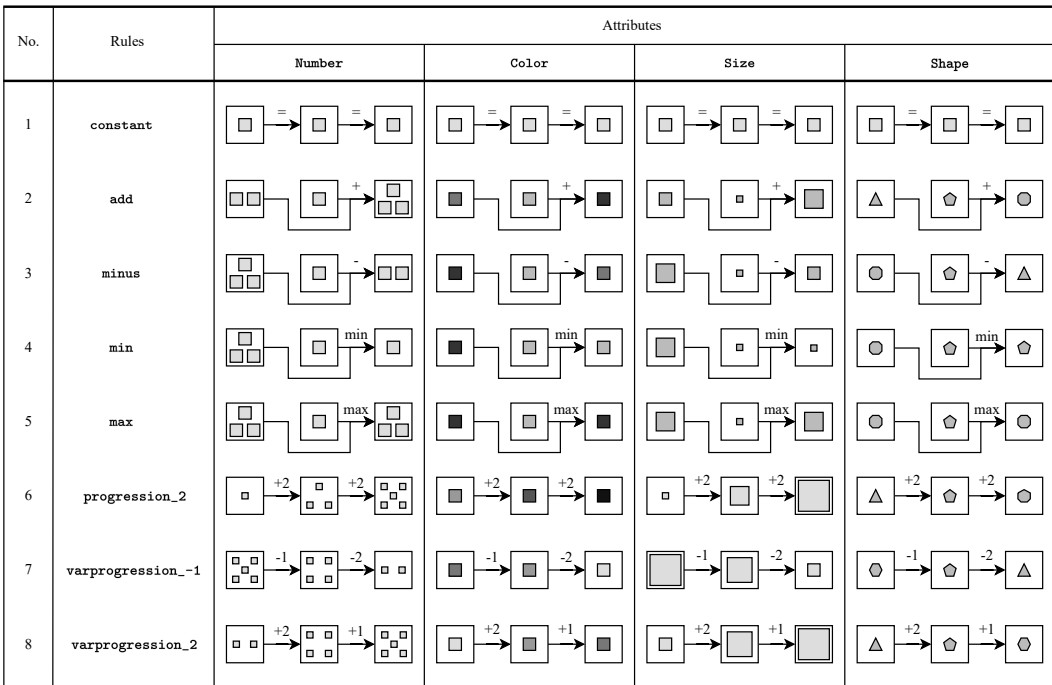

Figure 7: The embodiment of all 8 rules involved in the joint training stage on the 4 attributes, i.e., (Number, Color, Size, Shape).

In the speaker pre-training stage, the rules involved in each attribute are shown in Figure 8, which do not overlap with those in Figure 7, and have similar semantics to rules No.6, No.7, and No.8 in Figure 7. Therefore, pre-training the speaker with these rules does not provide agents with prior knowledge about the rules in the joint training stage, which only helps the speaker acquire reasoning capabilities to cope with the drifting context.

## A.2 A RPM case in rule-RAVEN

Figure 9 shows one RPM case of the rule-RAVEN dataset used in our experiments. The form of this case is consistent with Figure2. Figure2 is just for the convenience of illustration, only 1 attribute of the rule is drawn, but the actual rule consists of 4 attributes, i.e., (Number, Color, Size, Shape).

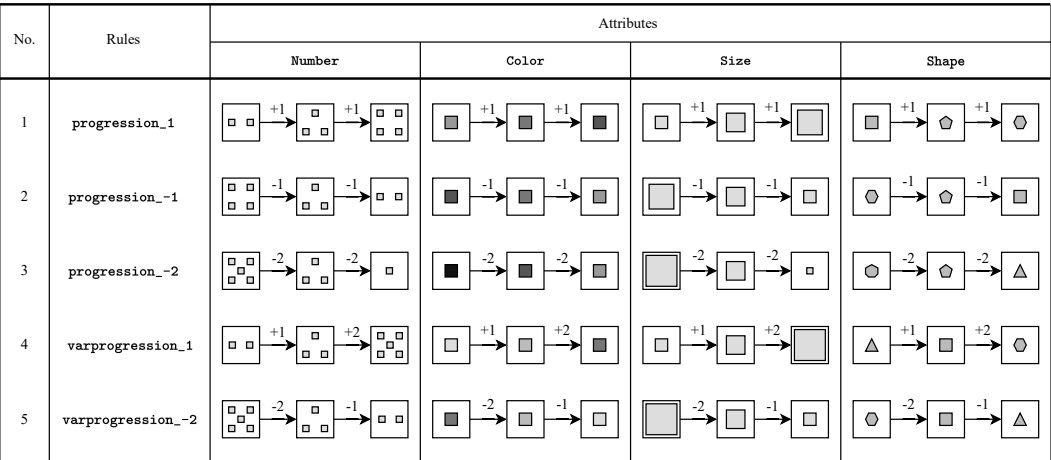

Figure 8: The embodiment of 4 extra rules involved in the speaker pre-training stage on the 4 attributes, i.e., (Number, Color, Size, Shape). These rules involved in the pre-training phase do not overlap with rules in the joint training stage.

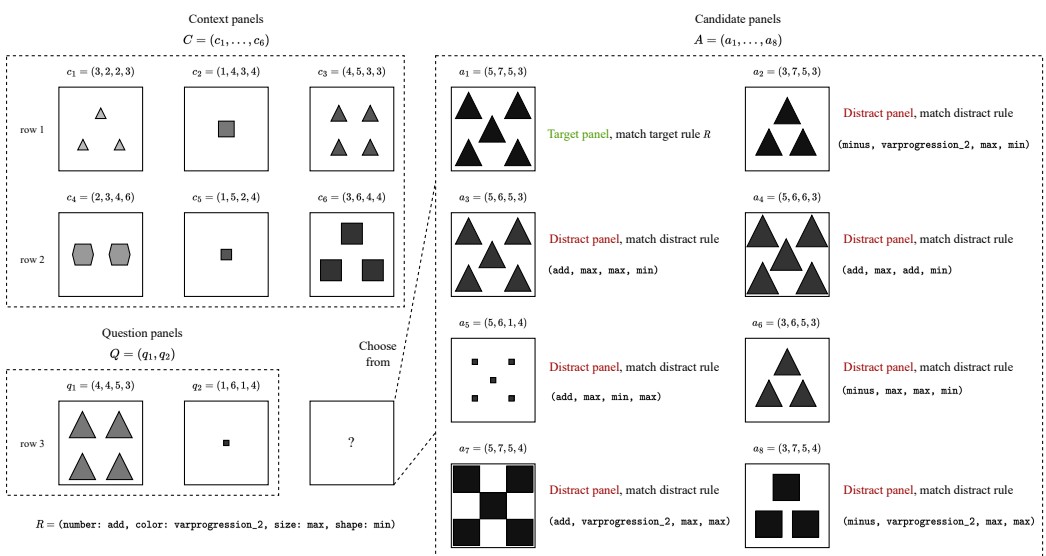

Figure 9: An example problem of the rule-RAVEN dataset with 4 attributes (i.e., (Number, Color, Size, Shape)).

### A.3 Details of generalization data splits

Figure 10 shows an example of four levels of generalization data splits described in Section 4.2.

For ID, `Inpo-ood`, and `Expo-ood-L2` generalization levels, we first get $7^4 = 2401$ rule combinations based on rules No.1 to 7 in Figure 7 on all 4 attributes. We then randomly sample 300 rule combinations from these 2401 rule combinations and generate 10 problems for each rule combination as the `Inpo-ood` data split (with a total of $300 \times 10 = 3000$ problems). For each of the remaining $7^4 - 300 = 2101$ rule combinations, we generate 10 problems as the train data split (for ID, `Inpo-ood`, and `Expo-ood-L2`) and 10 problems as the ID data split (with a total of $2101 \times 10 = 21010$ problems for training and 21010 for ID). We finally combine rule No.8, excluded in the previous process, with the rules on other attributes to get $8^4 - 7^4 = 1695$ rule combinations and generate 10 problems for each rule combination as `Expo-ood-L2` data split (with a total of $1695 \times 10 = 16950$ problems).

For the `Expo-ood-L1` generalization level, we exclude rule No.6 on attribute 1, rule No.3 on attribute 2, rule No.4 on attribute 3, and rule No.8 on attribute 4, getting 2401 rule combinations remaining. By comparing the accuracy difference between `Expo-ood-L1` and `Expo-ood-L2` levels (both `Expo-ood-L1` and `Expo-ood-L2` exclude one rule on each attribute, but on `Expo-ood-L1` level, rules excluded on one attribute appear on other attributes), we can check whether the language describing the rules can help the listener apply the rules across attributes. To compare fairly with `Expo-ood-L2` level, we randomly exclude 300 rule combinations and generate 10 problems for each of the remaining 2101 rule combinations as the training set for `Expo-ood-L1` level (with a total of $2101 \times 10 = 21010$ problems). We combine the rules excluded in the previous process with the rules on other attributes to get $8^4 - 7^4 = 1695$ rule combinations and generate 10 problems for each rule combination as `Expo-ood-L1` data split (with a total of $1695 \times 10 = 16950$ problems).

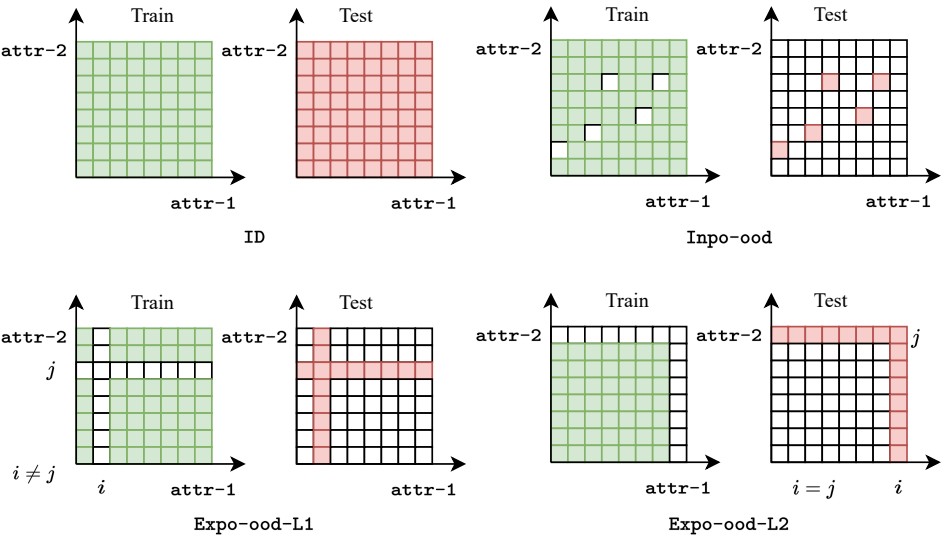

Figure 10: An example (with 2 attributes, (`attr-1`, `attr-2`)) to illustrate 4 generalization levels.

## B  Details of Agent Model

The model diagram of the speaker and listener is shown in Figure 11. Some hyperparameters of the model are listed in Table 4. Please refer to the source code in the supplementary materials for more implementation details.

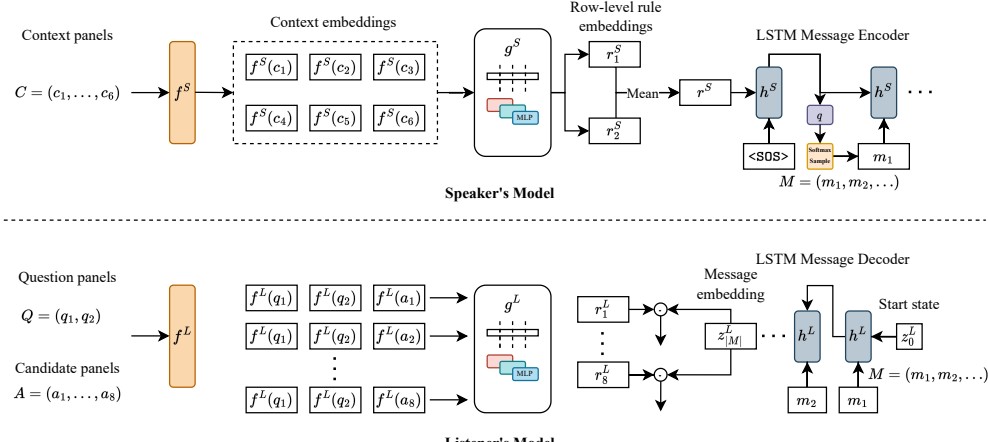

Figure 11: The Speaker's and Listener's Model.

Table 4: Some hyperparameters of the model.

| Hyperparameter | Value of $N \in \{20, 30, 40, 80\}$ |
| --- | --- |
| $f^S$ and $f^L$ output dim | $[80, 120, 160, 240]$ |
| Groups of $g^S$ and $g^L$ | $[80, 120, 160, 240]$ |
| Experts of $g^S$ and $g^L$ | 5 |
| $g^S$ and $g^L$ output dim | $[400, 600, 800, 1200]$ |
| $h^S$ and $h^L$ hidden dim | $[400, 600, 800, 1200]$ |

## C  Results of Diverse Language Sizes

The proposed two-stage training method does not introduce constraints on the size of the emerged language (i.e., `message_length` and `vocabulary_size`). The reason is that the speaker updates $f^S$, $g^S$, and $h^S$ in the first stage of training, while only load parameters of $f^S$, $g^S$ (without $h^S$) in the second stage. Therefore, even if the $\mathbb{1}(r_i, m_i)$ operator on message encoder $h^S$ constrains the equal size between message $m_i$ and rule $r_i$ in the first stage, we can the training a new $h^S$ with reconfigurable language size in the second stage (joint) training.

We also provide results with diverse language sizes (Table 5) in our reasoning game and get similarly high generalization performance (accuracy $\sim 0.95$).

Table 5:  Generalization accuracy with different language sizes (`message_length` $M$, `vocabulary_size` $V$) and attribute values $N$.

| $(M, V, N)$ | ID | Inpo-ood |
| --- | --- | --- |
| $(6, 15, 20)$ | $0.9539 \pm 0.0018$ | $0.9528 \pm 0.0015$ |
| $(6, 30, 20)$ | $0.9522 \pm 0.0015$ | $0.9513 \pm 0.0031$ |
| $(6, 15, 30)$ | $0.9429 \pm 0.0053$ | $0.9385 \pm 0.0045$ |
| $(6, 30, 30)$ | $0.9411 \pm 0.0022$ | $0.9362 \pm 0.0014$ |

# D    Token-level Analysis of the Emerged Language

We compute the probability distribution $P(message|attribute, rule)$ of a randomly selected seed and provide the most probable tokens for a given attribute and rule, as shown in Table 6.

Table 6: Given attributes and rules, the most probable tokens at each position.

| Rules | Color | Number | Size | Shape |
|---|---|---|---|---|
| add | **K**, C, K, H | B, C, B, C | **K**, L, K, K | B, K, B, C |
| minus | **B**, L, B, B | K, O, G, O | **G**, G, G, H | J, J, J, J |
| min | **K**, C, K, C | J, L, J, C | **K**, O, K, K | B, B, B, C |
| max | **B**, B, B, K | K, O, K, K | **K**, G, J, B | F, B, K, C |
| constant | **K**, B, K, K | B, K, C, K | **K**, C, K, C | K, B, K, C |
| progression_2 | **B**, L, B, B | K, O, K, O | **G**, G, H, H | J, O, J, J |
| varprogression_-1 | **K**, C, K, **C** | B, J, J, **C** | **K**, O, K, **C** | B, K, B, **C** |

The results indicate that tokens exhibit regular patterns (i.e., language-like syntax and compositionality) for different attributes and rules. For example, almost all rules related to attribute 'color' start with tokens 'K' or 'B', and attribute 'size' start with tokens 'K' or 'G'. On another dimension, the rule 'varprogression_-1' across all attributes ends with token 'C'.

