# OpenReview forum: "Emergent Communication for Rules Reasoning"
_NeurIPS.cc/2023/Conference — NeurIPS 2023 poster_

### Official Review · Reviewer_9tUw · 2023-07-03

**Soundness:** 3 good
**Presentation:** 4 excellent
**Contribution:** 2 fair
**Rating:** 6
**Confidence:** 4

**Summary:**

This work investigates the emergent communication framework for reasoning rules. That is, unlike prior studies that focus on communication about perceived low-level contexts, this paper proposes a cognition-oriented environment to encourage agents to reason and communicate about high level-rules. To this end, it introduces a new interesting and unbiased benchmark, rule-RAVEN. This benchmark, as opposed to the original one (I-RAVEN) avoids overfitting and pushes the agents to have an actual communication protocol.

The authors show, with different experiments, that agents are able to succeed in the reasoning tasks and develop a compositional and semantically stable language.

**Strengths:**

The authors introduce a well-thought benchmark that could be beneficial for future works to analyze the content of emergent languages. This benchmark, which is a modification of I-RAVEN, forces agents to develop an actual communication protocol. They perform the needed ablation to show its benefit compared to I-RAVEN.

Furthermore, this paper is well-written, and a detailed description of the setting, and hyper-parameters are provided (on top of the code).

**Weaknesses:**

The main weakness of this work is its motivation. As stated in the paper, the goal of the emergent communication framework is to:
- either study the origin of the human languages
and/or
- develop intelligent communicating artificial agents

It is unclear what this work's position is. If the former, is there a theory that our language emerged to communicate about a high-level reasoning task? If so, can this line of work be clarified in the paper? If the goal is to develop communicating agents, communicating about visual inputs is more practical for human-agent interactions.

**Questions:**

Can you explain further the experiments of the paragraph "Rule-RAVEN dataset"  (line 260)? In particular, I don't understand why we have an "unsuccessful" communication game with "rule-RAVEN" dataset if the speaker was already trained (in a two-stage setting). That is, in this training regime, the speaker's language is not random, and the listener should be able to succeed in the communication game (or at least have a good enough accuracy) without modifying the speaker's language). Maybe you can elaborate more on the stage 1 training?

Also, you state that "for the listener’s train accuracy still achieves ~0.9 even if the speaker’s message is completely ignored". How do you check if the speaker's message is completely ignored?

**Limitations:**

.

---

> ### Author Rebuttal · Authors · 2023-08-09
>
> Thanks for the thoughtful comments. We would like to clarify the concerns as follows:
>
> 1. > Clarify this work's position: study the origin of the human languages or develop intelligent communicating artificial agents.
>
>
>
>    Our work's position closer to the later, i.e., developing intelligent communicating artificial agents, cause we mainly focus on verifying the generalizable and transferable ability of agents in this paper.
>
>
>
>    In addition,  now we supplement experiments for demonstrating the emerged language's transferability performance on image-based downstream real-world tasks. Specifically, we first generate symbolic data with $Number \in \\{1, \dots 9\\}$, $Color \in \\{1, \dots 9\\}$, $Shape \in \\{3, \dots 9\\}$ (triangle, square, ..., nonagon), and $Size \in \\{1, \dots, 9\\}$ using the rule-RAVEN dataset. We then implemented a render to draw the panel's symbol as a $320\times320$ grayscale image. Finally, we train new listeners using the message from the symbolic environment and question-candidate panel images (we replace $f^L$ with a 5-layer ConvNet to process the image input). After 20 epochs of training, the training accuracy of the listener is in Table 1.
>
>
>
>    **Table 1**: Transfer accuracy of language emerged on symbolic reasoning task to image-based downstream reasoning tasks ('agent': language from a well-trained speaker; random:  a random language).
>
>    | Attribute values |        agent        |       random        |
>    | :--------------: | :-----------------: | :-----------------: |
>    |        20        | $0.9193 \pm 0.0029$ | $0.3519 \pm 0.0245$ |
>    |        30        | $0.9168 \pm 0.0084$ | $0.3030 \pm 0.0182$ |
>    |        40        | $0.8996 \pm 0.0109$ | $0.3406 \pm 0.0618$ |
>
>    The results show that languages emerging in symbolic environments can transfer (with ~ 0.9 accuracy) to downstream visual reasoning tasks.
>
>
>
>    Moreover, for the former (i.e., study the origin of the human languages), cognitive literature [1] provides the theory:
>
>    > *The ability to use linguistic signs to express freely-formed thoughts marks "the true distinction between man and animal" or machine.*
>
>     We will clarify in the next version of the paper.
>
>
>
>    Refs:
>
>    [1] Chomsky, N. (2004). "Chapter 15 Language and Mind: Current Thoughts on Ancient Problems".
>
>
>
> 2. > Clarify the paragraph 'Rule-RAVEN dataset' at line 260.
>
>    This paragraph compares the impact of different panel generation methods in two datasets (i.e., rule-RAVEN and I-RAVEN) on agents' overfitting.
>
>    Specifically, an undeniable fact is that, in a message-blocked scene (where the speaker is bypassed, and the message is always a constant, making it unusable by the listener), due to the complete ineffectiveness of the message, a higher accuracy (i.e., the likelihood of the listener correctly selecting the target) implies a greater degree of inductive bias, which in turn suggests a larger overfitting risk.
>
>    In this paragraph, based on the experiments under such message-blocked scenes, we prove that the candidate panels generated by the I-RAVEN-style method lead to a more severe overfitting, i.e., the listener achieves ~0.9 accuracies by just simply analyzing the question-candidate answer panels without any information from the speaker (lines with the legend 'I-RAVEN_20/30/40' in Figure 4, significantly higher than rule-RAVEN.)
>
>    We will revise this paragraph to make it clearer and less confusing.
>
>
>
>
>
> 3. > How do you check if the speaker's message is completely ignored?
>
>    The message-blocked scenes (where the speaker is bypassed, and the message is always a constant, making it unusable by the listener) are equivalent to the listener completely ignoring the message from the speaker, which is the lower bound of the accuracy of the communication task.

---

### Official Review · Reviewer_M8eN · 2023-07-03

**Soundness:** 3 good
**Presentation:** 4 excellent
**Contribution:** 3 good
**Rating:** 7
**Confidence:** 4

**Summary:**

This paper takes the ever-popular Lewis signalling game for emergent
communication and studies experiments on rule-focused communication as opposed
to the perception-focused communication of prior work.  In particular, it uses
a modified version of Raven's progressive matrices to formulate a signalling
game directly on attribute-value vectors which requires pattern
recognition/reasoning to complete.  The authors find that agents can learn to
communicate when using a two-stage curriculum which essentially pretrains the
speaker for more stable communication at the beginning of the communication
stage.  The experiments demonstrate that the resulting emergent language
correlates better with the underlying rules of observations rather than the
individual observations themselves.


**Strengths:**

## Originality
- `[major]` Addresses the signalling game from a new perspective, i.e.,
  reasoning instead of perception.
- `[minor]` Introduces a new dataset.
## Quality
- `[major]` Presents good variety of empirical evaluation with clear results
- `[minor]` Presents different levels and senses of "generalization".
## Clarity
- `[minor]` Details for implementation are presented without being overwhelming
## Significance
*See Originality.*


**Weaknesses:**

The paper is relatively complete, but what keeps my rating from being higher is
that there is relatively sparse comparison with prior work.  Such comparison
would better contextualize the results and increase its significance.  For
further details, see the *Questions* section of the review.


**Questions:**

- How does this work compare with prior art which also uses categorical
  variables for a signalling game, even if there is no reasoning element to the
  game?
- What are the particular effects on an emergent language of
  a reasoning-focused game versus one that is perception-focused?  What sort of
  inductive biases are present in the task that has a downstream effect on the
  language?

## Minor Comments/Questions
- `Line 46` "inner": typo?
- `Line 46, 86` "inter": typo?
- `Line 122` Do not use curly braces for ordered sequences; use parentheses.
- `Paragaph @ 119` Why is it necessary for there to be no ambiguity?  This sort of ambiguity shows up frequently in human communication.
- `Line 251` "cause" -> "because"
- `Paragaph @ 246` I would use "shows" or "demonstrates" instead of "proves" since it is not a formal, mathematical proof.
- Use the default LaTeX placement of tables/figures instead of `[h]`; the former is less distracting.


**Limitations:**

N/A.

---

> ### Author Rebuttal · Authors · 2023-08-09
>
> Thanks for the thoughtful comments. We would like to clarify the concerns as follows:
>
> 1. > Comparison with previous work using symbolic dataset.
>
>    We compare with previous emergent language work using symbolic datasets  (cited in our paper) from multiple angles: research goals, task orientation, input complexity, and agent training.
>
>
>
>    **Table 1**: Comparison of previous emergent language work using symbolic datasets.
>
>    |                  | Ours                                                         | Others                                                       |
>    | ---------------- | ------------------------------------------------------------ | ------------------------------------------------------------ |
>    | Research Goals   | Emergent cognition-based communication for rules reasoning.  | Efficiency of Language Coding [1], Ease-of-teaching [2], population heterogeneity [3], and promote structured, generalizable languages [4]. |
>    | Task Orientation | Extracting and using **inter-context** rules to handle reasoning problems. | Identifying the **inner-context** object attributes for discrimination or reconstruction task [1-4]. |
>    | Input Complexity | Multiple symbolic vectors with attributes.                   | One symbolic vector with attributes [1-4].                   |
>    | Agent Training   | Two-stage training due to the contexts-and-semantics bilaterally drifting task. | End-to-end training due to the training simplistic task [1-3], iterated training [4]. |
>
>    We would supplement more related work and revise our paper.
>
>
>
>    Refs:
>
>    [1] Chaabouni, R., Kharitonov, E., Dupoux, E., & Baroni, M. (2019). Anti-efficient encoding in emergent communication.
>
>    [2] Li, F., & Bowling, M. (2019). Ease-of-teaching and language structure from emergent communication.
>
>    [3] Rita, M., Strub, F., Grill, J. B., Pietquin, O., & Dupoux, E. (2022). On the role of population heterogeneity in emergent communication.
>
>    [4] Ren, Y., Guo, S., Labeau, M., Cohen, S. B., & Kirby, S. (2020). Compositional languages emerge in a neural iterated learning model.
>
>
>
> 2. > Semantic effects and inductive biases of a reasoning-focused game versus one that is perception-focused.
>
>
>
>    **Table 2**: Comparison of reasoning-focused and perception-focused games from both perspectives of semantic effects and inductive biases.
>
>    |                      | Reasoning-focused game (ours)                                | Perception-focused game                                      |
>    | -------------------- | ------------------------------------------------------------ | ------------------------------------------------------------ |
>    | **Semantic effects** | The emerged language describes changing patterns (i.e., rules) between *multiple* input contexts. | The emerged language describes perceptual features of objects (or attributes) within a *single* input context. |
>    | **Inductive biases** | The game settings force agents to cooperative reason by conveying abstract rules *implicit* in contexts. | The game settings force agents to discriminate or reconstruct targets by conveying perceptual features of contexts. |
>
>
>
> 3. > `Paragraph @ 119` Why is it necessary for there to be no ambiguity?
>
>
>
>    From the listener's perspective, structural requirements (unambiguous) are necessary. Specifically, when considering a dataset comprising multiple RPM problems, the occurrence of ambiguity (i.e., attribute values and rules are not one-to-one correspondence) would result in an imbalanced *prior probability distribution* of rules' frequency in the dataset. This imbalance implies the inequality significance between rules, inducing inductive bias in the dataset, thus posing a higher risk of listener's overfitting.
>
>
>
> 4. > About other minor questions.
>
>
>
>    We will revise the text format and typos, and modify inappropriate expressions in line 122 and paragraph at 246.

---

> > ### Comment · Reviewer_M8eN · 2023-08-11
> >
> > I have read the author's rebuttal.
> > I think the proposed table would be a great addition to the paper, although I do not think it would go quite far enough to get me to increase my score above a 7/10 (there would need to be empirical ablation studies), but I think the paper is largely adequate with the proposed changes.
> >
> > Minor edit: "inner-context" -> "intra-context"

---

### Official Review · Reviewer_Ntdv · 2023-07-05

**Soundness:** 3 good
**Presentation:** 3 good
**Contribution:** 3 good
**Rating:** 6
**Confidence:** 4

**Summary:**

This paper proposed a new environment along with the training framework for emergent communication of abstract rules. They designed a context generation pipeline rule-RAVEN to avoid overfitting and a two-stage curriculum training method for more stable convergence. They evaluated the emerged language from the perspectives of generalization and transfer learning.


**Strengths:**

1. This paper proposed a new research angle of abstract rule reasoning for emergent communication. The context requires the agent to go beyond the low-level perceptual features and communicate more abstract rules.
2. The candidate pool is smartly designed to motivate agents to extract rules from the context.
3. A suite of comprehensive evaluations is designed to measure the generalization of the emerged languages.


**Weaknesses:**

1. Structural requirements of the new benchmark may need to be further explained: I am not clear about why the rules must be unambiguous. From my understanding, though the multiple rules can be applied to the current context, as long as the agents can communicate either of the rules, the receiver should capture the correct candidate? Though Figure 4 demonstrates the receiver can select the candidate correctly without the sender’s message trained with rule-RAVEN, further experiments/explanations are still needed to show that:

    a. The communication training benefits from the structural or functional requirement or both.

    b. The language that emerged using the I-RAVEN dataset is not/less generalizable/compositional/transferable.

2. Sender’s rule reasoning and perception encoding are entangled. In the first stage, both $g^S$ and $f^S$ are trained. Though the training data is not shown in the communication stage, it will still introduce structural information because of the term $\mathbb{1}(r_i, m_i)$.

    a. does that require the length of the messages to equal the size of the rules?


**Questions:**

In Sec 5.1, how distance(rule) and distance(panel) are computed? Just want to clarify whether these two distances are comparable.



**Limitations:**

1. Though the goal of the task is to emerge the language for abstract rules, it will also be interesting to know whether the receiver can learn to induce rules after the communication (instead of applying rules during the communication, not required experiments). Similar to ETL, you can test the accuracy of the reasoning problem on the communicated receiver without further training.
2. As the author mentioned, the current context input is a structured symbol. It will strongly encourage compositional language. It will be interesting to know how agents can emerge languages in a raw pixel input.
3. Can the emerged languages generalize to contexts with different attributes?

---

> ### Author Rebuttal · Authors · 2023-08-09
>
> Thanks for the thoughtful comments. We would like to clarify the concerns as follows:
>
> 1. > Experiments/explanations about 1a and 1b.
>
>
>
>    (1a) The communication training process benefits from both the structural and functional requirements.
>
>
>
>    From the listener's perspective, structural requirements (unambiguous) are necessary. Specifically, when considering a dataset comprising multiple RPM problems, the occurrence of ambiguity (i.e., attribute values and rules are not one-to-one correspondence) would result in an imbalanced *prior probability distribution* of rules' frequency in the dataset. This imbalance implies the inequality significance between rules, inducing inductive bias in the dataset, thus posing a higher risk of listener's overfitting.
>
>
>
>    From the perspective of the speaker, functional requirements are also necessary. Specifically, our rule-based candidate-panel generation algorithm deliberately confuses target rules between question-candidate panel pairs. Such deliberate confusion forces the speaker to initiate meaningful communication to help the listener.
>
>
>
>    (1b) The language that emerged using the I-RAVEN dataset is not generalizable, transferable, and compositional.
>
>    For generalizable: We first emerge language using the I-RAVEN dataset, then test the language effectiveness on the rule-RAVEN dataset (these two datasets share the same rules but differ in how they generate candidate panels). Experimental results (Table 1) show low generalization performance of the language (i.e., ID/Inpo-ood accuracy only ~0.6, close to the message-blocked scene).
>
>
>
>    **Table 1**: Generalization accuracy with different attribute values N.
>
>    |  N   |         ID          |      Inpo-ood       | Message-blocked scene |
>    | :--: | :-----------------: | :-----------------: | :-------------------: |
>    |  20  | $0.6312 \pm 0.0017$ | $0.6270 \pm 0.0024$ |  $0.6220 \pm 0.0019$  |
>    |  30  | $0.6271 \pm 0.0034$ | $0.6247 \pm 0.0025$ |  $0.6215 \pm 0.0033$  |
>
>
>
>    For transferable: The experimental results (Table 2) also show low language transfer performance (accuracy only ~0.6, close to random).
>
>
>
>    **Table 2**: Transfer accuracy of (source S, target T) attribute values ('agent': language from a well-trained speaker; random:  a random language).
>
>    |  (S, T)  |        agent        |       random        |
>    | :------: | :-----------------: | :-----------------: |
>    | (20, 30) | $0.6262 \pm 0.0057$ | $0.6255 \pm 0.0036$ |
>    | (20, 40) | $0.6246 \pm 0.0002$ | $0.6248 \pm 0.0025$ |
>    | (30, 40) | $0.6236 \pm 0.0007$ | $0.6209 \pm 0.0034$ |
>
>
>
>    For compositional: Further analysis of the language shows that, when using I-RAVEN, the speakers describe the rules of different reasoning problems with constant messages (the listener completely ignores the speaker's message) on all random seeds, which indicates that the language is not compositional.
>
>
>
> 2. > Does the proposed training method have a limit on the message size?
>
>
>
>    The training method does not introduce constraints on the length of messages. Specifically, the speaker updates  $f^S$, $g^S$, and $h^S$ in the first stage of training, while only load parameters of $f^S$, $g^S$ (without $h^S$) in the second stage. Therefore, even if the $1(r_i,m_i)$ operator on message encoder $h^S$ constrains the equal size between message $m_i$ and rule $r_i$ in the first stage, we can the training a new $h^S$ with reconfigurable message size in the second stage (joint) training.
>
>
>
>    To demonstrate this claim, we tried diverse message sizes (Table 3) in our reasoning game, and get similarly high generalization performance (accuracy ~0.95).
>
>
>
>    **Table 3**: Generalization accuracy with different message sizes (message length M, vocabulary size V) and attribute values N.
>
>    |  (M, V, N)  |         ID          |      Inpo-ood       |
>    | :---------: | :-----------------: | :-----------------: |
>    | (6, 15, 20) | $0.9539 \pm 0.0018$ | $0.9528 \pm 0.0015$ |
>    | (6, 30 ,20) | $0.9522 \pm 0.0015$ | $0.9513 \pm 0.0031$ |
>    | (6, 15, 30) | $0.9429 \pm 0.0053$ | $0.9385 \pm 0.0045$ |
>    | (6, 30, 30) | $0.9411 \pm 0.0022$ | $0.9362 \pm 0.0014$ |
>
>
>
> 3. > How distance of rules/panels are computed?
>
>
>
>    - For rules, different rules belong to different categories, so we first compute rule vectors one-hot encoding. Then, we use the cosine distance (in this case, cosine distance is similar to the normalized hamming distance) to calculate the distance between rules.
>    - For panels, according to 4 attributes（number, shape, color, size）value, each context panel can be represented as a 4-dim integer vector. Within one RPM problem, we directly concatenate all 6 context panels into a 24-dim integer vector. Then, we use the cosine distance to calculate the distance between context panels of RPM problems.
>
>
>
> 4. > Whether the listener can learn to induce rules after the communication.
>
>
>
>    We infer that the listener can learn to induce rules after the communication because the key to cooperative reasoning is that the speaker and listener have a similar ability to reason and agree on the message's linguistics (rules mapping). We check this by testing the accuracy of the reasoning problem after exchanging the parameters of the reasoning modules ($g^S$ and $g^L$) of the well-trained speaker and listener.
>
>
>
> 5. >  Emerge language in a raw pixel input, and show compositionality.
>
>
>
>    We supplement the transfer performance of emerged language to image-based downstream reasoning task experiments in reviewer LuwD(Q1) and show the compositionality of emerged language in Reviewer 3wvB(Q2).
>
>
>
> 6. >  Languages generalize to contexts with different attributes.
>
>
>
>    The emerged language can generalize to contexts with different attributes as long as ensuring the quality of the representations produced by the perception module (no effect on the cognition module).

---

### Official Review · Reviewer_3wvB · 2023-07-06

**Soundness:** 3 good
**Presentation:** 3 good
**Contribution:** 3 good
**Rating:** 6
**Confidence:** 5

**Summary:**

This paper introduces a novel setting for abstract reasoning (i.e., RAVEN problems) by proposing a speaker-listener framework for communicating higher-level abstract rules. The authors propose an unbiased dataset (rule-RAVEN) to overcome overfitting in the original RAVEN-family datasets (I-RAVEN), and propose a two-stage curriculum agent training method for successful communication. Experiments have shown the efficacy of the curriculum training for solving the rule-RAVEN and out-of-distribution generalization.

**Strengths:**

- I like the idea of both: i). introducing communicative game settings to abstract reasoning tasks, and ii). see how higher-level relational abstractions (instead of low-level perceptual features) emerge in communicative games. The limited capacity communication channel formulation can lead to emergent abstractions for problem-solving, including more powerful representations for abstract reasoning and concept learning. Previous attempts in drawing are good cases but not complex enough to depict the importance of abstraction and emergent language. This preliminary trial on RAVEN tests sets a suitable problem formulation for emergent communication in abstract reasoning.

- The rule-RAVEN dataset effectively mitigates the existing bias in the I-RAVEN dataset, making the speaker-listener communication valid.

- The paper is well-written and easy to read. The flow of writing in section 5 is also appropriate for addressing potential concerns for readers.


**Weaknesses:**

Although I like the task settings in this paper, the experiment and proposed methods appear to have some weaknesses. I list them as follows:

- The communicative formulation is very similar to Mu & Goodman, 2021. It seems this work (communicative RAVEN) is a special case of generalization, shifting from learning object-centric, attribute-level concepts (e.g., shape red or blue) to learning relational concepts (number-increasing). Authors should address more comparisons to these existing formulations.

- The evaluation for emergent language is still quite limited. For example, can you probe the learned language to see if it can be linearly projected to some algebraic representations for relational concepts (e.g., the "number increasing" concept can be described as a multiplication matrix in Zhang et al., 2022) or just explicitly manipulate them and see if they have some language-like syntax or compositionality emerged.

- The use of symbolic RAVEN and two-stage curriculum training (with the first stage supervised learned) made me doubt the applicability of this communicative method to more complex or real-world tasks. For example, Mu & Goodman, 2021 used a real-world dataset, pixel input, and end-to-end training.

refs:
1. Mu, J., & Goodman, N. (2021). Emergent communication of generalizations. Advances in Neural Information Processing Systems, 34, 17994-18007.

2. Zhang, C., Xie, S., Jia, B., Wu, Y. N., Zhu, S. C., & Zhu, Y. (2022, October). Learning algebraic representation for systematic generalization in abstract reasoning. In European Conference on Computer Vision (pp. 692-709). Cham: Springer Nature Switzerland.

**Questions:**

See the weakness section. I am open to changing my score, so I hope the authors can address these concerns.

**Limitations:**

The authors did not address the limitations.

---

> ### Author Rebuttal · Authors · 2023-08-09
>
> Thanks for the thoughtful comments. We would like to clarify the concerns as follows:
>
> 1. > More comparisons to existing formulations [1].
>
>
>
>    Similar to many previous works, the agent communicative formulation (i.e., the definition of 'abstract concepts') in [1] identifies the **inner-context** object attributes (e.g., shape, color). While in our RPM-based reasoning game, the agent communicative formulation is extracting and using **inter-context** rules to handle reasoning problems. We will classify previous work based on different communicative formulations and revise the related work in our paper.
>
>
>
> 2. > Explore if the emergent language has some language-like syntax or compositionality.
>
>
>
>    Inspired by [2], we compute the probability distribution $P(token\_sequence|attribute, rule)$ under randomly selected seeds and give the most probable tokens for a given attribute and rule, as shown in Table 1.
>
>    **Table 1**: Given attributes and rules, the most probable tokens at each position.
>
>    |       Rules       |       color        | number         | size               | shape          |
>    | :---------------: | :----------------: | -------------- | ------------------ | -------------- |
>    |        add        |   **K**, C, K, H   | B, C, B, C     | **K**, L, K, K     | B, K, B, C     |
>    |       minus       |   **B**, L, B, B   | K, O, G, O     | **G**, G, G, H     | J, J, J, J     |
>    |        min        |   **K**, C, K, C   | J, L, J, C     | **K**, O, K, K     | B, B, B, C     |
>    |        max        |   **B**, B, B, K   | K, O, K, K     | **K**, G, J, B     | F, B, K, C     |
>    |     constant      |   **K**, B, K, K   | B, K, C, K     | **K**, C, K, C     | K, B, K, C     |
>    |   progression_2   |   **B**, L, B, B   | K, O, K, O     | **G**, G, H, H     | J, O, J, J     |
>    | varprogression_-1 | **K**, C, K, **C** | B, J, J, **C** | **K**, O, K, **C** | B, K, B, **C** |
>
>    The results indicate that tokens exhibit regular patterns (i.e., language-like syntax and compositionality) for different attributes and rules. For example, almost all rules related to attribute 'color' start with tokens 'K' and 'B', and attribute 'size' start with tokens 'K' and 'G'. On another dimension, the rule 'varprogression_-1' across all attributes ends with token 'C'.
>
>
>
> 3. > Apply the two-stage curriculum training method to real-world tasks [1] with pixel input, and verify its effectiveness.
>
>
>
>    First, to verify that our two-stage training method can be applied to real-world datasets, we demonstrate the transferability of language produced by two-stage trained agents on image-based (i.e., pixel input) downstream tasks. Specifically,  we firstly generate symbolic data with $Number \in \\{1, \dots 9\\}$, $Color \in \\{1, \dots 9\\}$, $Shape \in \\{3, \dots 9\\}$ (triangle, square, ..., nonagon), and $Size \in \\{1, \dots, 9\\}$ using the rule-RAVEN dataset. We then implemented a render to draw the panel's symbol as a $320\times320$ grayscale image. Finally, we train new listeners using the message from the symbolic environment and question-candidate panel images (we replace $f^L$ with a 5-layer ConvNet to process the image input). After 20 epochs of training, the training accuracy of the listener is in Table 2.
>
>
>
>    **Table 2**: Transfer accuracy of language emerged on symbolic reasoning task to image-based downstream reasoning tasks ('Agent' represents using the language emerged by a well-trained speaker, and 'random' represents using a random language).
>
>    | Attribute values |        agent        |       random        |
>    | :--------------: | :-----------------: | :-----------------: |
>    |        20        | $0.9193 \pm 0.0029$ | $0.3519 \pm 0.0245$ |
>    |        30        | $0.9168 \pm 0.0084$ | $0.3030 \pm 0.0182$ |
>    |        40        | $0.8996 \pm 0.0109$ | $0.3406 \pm 0.0618$ |
>
>    The results show that languages emerging in symbolic environments can transfer (with ~ 0.9 accuracy) to downstream visual reasoning tasks.
>
>
>
>    Second, we qualitatively analyze why the two-stage training method (especially the first stage) is still effective for real-world datasets. The first stage (i.e., supervised training) aims to warm up the speaker, enabling it to generate higher-quality messages during early epochs. This stage is crucial in preventing communication from getting trapped in a local optimum. For instance, in our work, we aim to avoid situations where the listener completely disregards the speaker’s message or where the speaker only conveys partial information.
>
>
>
> 4. > About limitation.
>
>    We summarized the limitation of our works as follow:
>
>    - Our work only focuses on language emergence on a clean symbolic-based reasoning dataset, lacking the exploration of more realistic stimuli-based (e.g., synthetic or natural images) reasoning datasets. We supplemented the transfer performance in image-based downstream reasoning tasks during the rebuttal (related results in reviewer LuwD Q1), and further investigation of the language emergence of image stimuli is still needed.
>
>    - The reasoning task (RAVEN) adopted in our work only requires the agent to complete the reasoning via a single round of interaction, simplifying the natural reasoning process.
>
>    - Our work only analyzes the semantics of the emerged languages at the message level, lacking fine-grained structural (gramma) and semantic analysis at the token level. We supplemented with coarse-grained token-level semantic analysis experiments during rebuttal (related results in Q2), and we would do more systematic analyzes further (e.g., the similarity of languages when given attributes and rules, and the degree of polysemy and ambiguity between tokens).
>
> Refs:
>
> [1] Mu, J., & Goodman, N. (2021). Emergent communication of generalizations.
>
> [2] Zhang, C., Xie, S., Jia, B., Wu, Y. N., Zhu, S. C., & Zhu, Y. (2022, October). Learning algebraic representation for systematic generalization in abstract reasoning.

---

> > ### Comment · Reviewer_3wvB · 2023-08-17
> > **Thanks**
> >
> > Thank the authors for the detailed response. I decide to increase my rating by one. I recommend acceptance of this paper.

---

### Official Review · Reviewer_LuwD · 2023-07-07

**Soundness:** 4 excellent
**Presentation:** 3 good
**Contribution:** 2 fair
**Rating:** 6
**Confidence:** 4

**Summary:**

This paper proposes an emergent communication game over abstract visual concepts, inspired by Raven's progressive matrices tests. The basic idea is to evaluate neural speakers and listeners on a communication game, where the speaker sees a collection of images encoding some abstract rule (e.g. "number of objects in the image is increasing"); the speaker must then generate a message that allows a listener to complete an unseen sequence. The authors show that agents trained to play this game indeed seem to learn to communicate the abstract rules for which they are trained for, as measured by intrinsic measures of language compositionality and ease of transfer to harder tasks.

**Strengths:**

- This is an interesting dataset and interesting problem in emergent communication which may be useful to the community. It indeed explores more abstract visual concepts than in existing work (though note that novelty over the existing EC literature is overclaimed; see Weaknesses).
- Careful controls for dataset difficulty (ensuring one distinct feature that can be used to solve each task; ensuring "hard negative" rules) show the authors' care to making sure this is a well-constructed dataset, including an analysis of to what extent existing
- Interesting experimental analysis shows that models seem to be (to some extent) communicating abstract rules, rather than superficial input features.

**Weaknesses:**

- Only a synthetic dataset consisting of clean symbolic inputs is evaluated. One could imagine more realistic settings requiring communication of rules at least over synthetic visual inputs, if not more realistic visual concepts. Similarly, there is no exploration of downstream transfer to other tasks that perhaps don't involve emergent communication, e.g. instruction following or visual reasoning. While this does not preclude publication, there's not a lot one can gain from this paper as it relates to actual realistic ML tasks. If the paper were to be rejected, IMO it would likely be because the experiments are just a little too synthetic/marginal to be useful to the broader NeurIPS community.
- The claim that existing work in EC does not at all care about expresing abstract generalizations or rules is a bit overblown. Separating inputs given to the student and teacher, so as to facilitate communication of abstract concepts, was introduced as early as Lazaridou (2017), recurs in Choi et al., Kiela et al., etc. Mu and Goodman (2022) also propose generalizations over abstract visual concepts involving multiple visual inputs, which is very similar to the task presented here. I do think the present work makes some interesting contributions over the existing literature, in that it is even more abstract, but the relation to existing work needs to be made more clear. Many of these papers are not discussed in detail and simply bucketed as  "forcing agents to descrie low-level features of images" (L31-32) which I believe is false. Section 2 Emergent Communication also completely neglects to discuss such efforts in the EC community.
- I think it's important for footnote 1 to be made more clear in the text, i.e. that this is not a grounded communication game over real images, despite many of the introductory figures seemingly suggesting this.

## Minor

- Title of paper and title on OpenReview do not match
- spaces between text and citations would be ideal
- It'd be interesting to see how pretraning agents on such visual reasoning communication tasks might improve performance on downstream visual reasoning tasks such as ARC (Chollet et al., ?)
- The description of the paragraphs in L119 and L128 as "structural" and "functional" requirements is a little confusing and nonstandard to me—it's not clear what structural and functional mean here. It might be appropriate for example to refer to the "functional requirement" as sampling "hard negative distractors", as is used in the terminology for contrastive learning for example. In other words, distractors should be sampled carefully so as to represent close but not quite correct rules that force the speaker and listener to communicate precisely the right rules.
- L171 "directly from the sketch" -> "directly from scratch"?

**Questions:**

- Did authors try varying the number of context panels given to the speaker? Or even show the partial sequence given to the listener? Wonder how this affects the languages' propensity to communicate abstractions; e.g. if the speaker sees the listener's partial sequence, does the language still communicate the abstract rule, or does the speaker internally learn the abstract rule but nevertheless convey the perceptual features (e.g. "single triangle")?

**Limitations:**

yes

---

> ### Author Rebuttal · Authors · 2023-08-09
>
> Thanks for the thoughtful comments. We would like to clarify the concerns as follows:
>
> 1. > Downstream transfer to visual reasoning tasks, and set more realistic environments requiring communication of rules over visual (image) inputs.
>
>
>
>    We demonstrate the emerged language's transfer performance on image-based downstream tasks. Specifically, we first generate symbolic data with $Number \in \\{1, \dots 9\\}$, $Color \in \\{1, \dots 9\\}$, $Shape \in \\{3, \dots 9\\}$ (triangle, square, ..., nonagon), and $Size \in \\{1, \dots, 9\\}$ using the rule-RAVEN dataset. We then implemented a render to draw the panel's symbol as a $320\times320$ grayscale image. Finally, we train new listeners using the message from the symbolic environment and question-candidate panel images (we replace $f^L$ with a 5-layer ConvNet to process the image input). After 20 epochs of training, the training accuracy of the listener is in Table 1.
>
>
>
>    **Table 1**: Transfer accuracy of language emerged on symbolic reasoning task to image-based downstream reasoning tasks ('agent': language from a well-trained speaker; random:  a random language).
>
>    | Attribute values |        agent        |       random        |
>    | :--------------: | :-----------------: | :-----------------: |
>    |        20        | $0.9193 \pm 0.0029$ | $0.3519 \pm 0.0245$ |
>    |        30        | $0.9168 \pm 0.0084$ | $0.3030 \pm 0.0182$ |
>    |        40        | $0.8996 \pm 0.0109$ | $0.3406 \pm 0.0618$ |
>
>    The results show that languages emerging in symbolic environments can transfer (with ~ 0.9 accuracy) to downstream visual reasoning tasks.
>
>
>
>    In fact, our symbolic rule-RAVEN dataset is sufficient to encourage agents to reason and communicate high-level rules. The reason is that, without changing the semantic information, the format of input data (e.g., visual or symbolic) only affects agents' perception but does not affect agents' cognition ability for rules reasoning.
>
>
>
>    Furthermore, we tend to investigate the emergence of language on more realistic (e.g., V-Prom [1]) and complex (e.g., ARC [2]) image-based reasoning datasets in future work.
>
>
>
>    Refs:
>
>    [1] Teney, D., & van den Hengel, A. (2020). V-prom: A benchmark for visual reasoning using visual progressive matrices.
>
>    [2] Chollet, F. (2019). On the measure of intelligence.
>
>
>
> 2. > Relation to existing work, which also 'communicate abstract concepts' (e.g., [3-6]), needs to be made more clear.
>
>
>
>    The original claim in related work does indeed lead to misunderstandings, and we do not deny that the previous work also 'communicate abstract concepts'. Based on the different definitions of "abstract concepts", we would do a more fine-grained comparison with previous work and revise the paper.
>
>    We take the several papers you mentioned ([3-6]) as examples:
>
>    - The 'abstract concepts' refers to inner-context object attributes (e.g., color, shape) in [3-5] or combinations of them (e.g., blue OR/AND triangle) in [6].
>    - While in our RPM-based reasoning game, the 'abstract concepts' refers to extracting and using inter-context rules to handle reasoning problems rather than selecting which object has specified attributes.
>
>    Refs:
>
>    [3] Lazaridou, A. (2016). Multi-agent cooperation and the emergence of (natural) language.
>
>    [4] Choi, E. (2018). Multi-agent compositional communication learning from raw visual input.
>
>    [5] Graesser. (2019). Emergent linguistic phenomena in multi-agent communication games.
>
>    [6] Mu, J., & Goodman, N. (2021). Emergent communication of generalizations.
>
>
>
> 3. > Footnote 1 (i.e., that this is not a grounded communication game over real images) needs to be made more clear in the text.
>
>
>
>    We point out that rule-RAVEN is a symbolic dataset in line 136 of the paper (last paragraph in section 3.2) and just give the reason in footnote 1. Thanks for pointing out misunderstandings that may arise here. We will more clearly illustrate that rule-RAVEN is a symbolic dataset and revise the paper.
>
>
>
> 4. > Try varying the number of context panels given to the speaker or listener, and research how the setting affects the languages' propensity to communicate abstractions.
>
>
>
>    We tried a new game setting: 1) the speaker receives the 6 context panels and 2 question panels, reasons the answer, and sends messages to the listener, and 2) the listener selects the target panel from the 8 candidate panels, only referring to the speaker's message.
>
>
>
>    Experimental results (Table 2) show that such a setting will lead to a slight decrease in generalization accuracy because this setting compels the speaker not only reasoning but also to describe the target panel accurately to the listener.
>
>
>
>    **Table 2**: Generalization accuracy on ID and Inpo-ood data splits.
>
>    | Attribute values |         ID          |      Inpo-ood       |
>    | :--------------: | :-----------------: | :-----------------: |
>    |        20        | $0.7273 \pm 0.0728$ | $0.7299 \pm 0.0725$ |
>    |        30        | $0.7996 \pm 0.0185$ | $0.7952 \pm 0.0172$ |
>
>    Moreover, there is evidence (Table 3) showing that such a setting leads the speaker tends to learn the abstract rule internally but convey the perceptual features of the target panel (Topsim(panel, message) > Topsim(rule, message)).
>
>
>
>    **Table 3**: Topsim between rules/target-panel and messages.
>
>    | Attribute values |    Rule-Message     |    Panel-Message    |
>    | :--------------: | :-----------------: | :-----------------: |
>    |        20        | $0.1776 \pm 0.0137$ | $0.2249 \pm 0.0683$ |
>    |        30        | $0.1571 \pm 0.0272$ | $0.2280 \pm 0.0306$ |
>
>
>
> 5. > About minor questions.
>
>
>
>    (1) Align to the title on the Openreview site, we will revise the paper's title.
>
>    (2) Real visual inputs related see the answer of Question 1.
>
>    (3) We will revise text format and typos, and improve writing in specified paragraphs (L119 and L128).

---

> > ### Comment · Reviewer_LuwD · 2023-08-15
> > **Thanks**
> >
> > Thanks to authors for their detailed response to my review, and for the follow up experiments which are quite interesting. Although I still think the task and domain are synthetic, I appreciate the inclusion of a more interesting downstream transfer task, and the author's rebuttal has solidified the difference between this work and related work. I'll increase my score to a 6.

---

### Decision · Program_Chairs · 2023-09-21

**Decision:**

Accept (poster)

**Comment:**

The paper presents an intriguing approach to emergent communication, focusing on reasoning and communication of high-level abstract rules using the rule-RAVEN benchmark. This novel perspective offers an interesting direction in the field, and the reviewers' assessments highlight both the strengths and potential areas for improvement of the paper.

The introduction of the rule-RAVEN benchmark, which encourages agents to develop communication protocols for abstract reasoning, is commended by the reviewers. The careful construction of the dataset and the two-stage curriculum training approach showcase the authors' attention to detail in experimental design. The exploration of higher-level rule-based reasoning through emergent communication aligns well with the broader goals of studying language emergence and developing intelligent communicating agents.

While the motivation behind the work is noted as a potential weakness, the reviewers appreciate the paper's contributions in introducing a well-designed benchmark and the detailed experimental analysis. Some reviewers have suggested a need for further comparisons with existing formulations, both perception-focused and reasoning-focused, to contextualize the results.

In conclusion, the innovative approach of reasoning-focused communication using the rule-RAVEN benchmark holds promise and presents a valuable contribution to the field. With some addressing of the reviewers' concerns, the paper should be accepted to share its insights and methodologies with the community.